# Cross-Resistance between Platinum-Based Chemotherapy and PARP Inhibitors in Castration-Resistant Prostate Cancer

**DOI:** 10.3390/cancers15102814

**Published:** 2023-05-18

**Authors:** Peter H. J. Slootbeek, Iris S. H. Kloots, Inge M. van Oort, Leonie I. Kroeze, Jack A. Schalken, Haiko J. Bloemendal, Niven Mehra

**Affiliations:** 1Department of Medical Oncology, Radboud University Medical Center, Geert Grooteplein-Zuid 10, 6525 GA Nijmegen, The Netherlands; 2Department of Urology, Radboud University Medical Center, Geert Grooteplein-Zuid 10, 6525 GA Nijmegen, The Netherlands; 3Department of Pathology, Radboud University Medical Center, Geert Grooteplein-Zuid 10, 6525 GA Nijmegen, The Netherlands

**Keywords:** BRCA2, castration-resistant prostate cancer, cross-resistance, DNA damage repair, homologous recombination, PARP inhibitors, platinum-based chemotherapy, precision medicine, precision oncology, prostate cancer

## Abstract

**Simple Summary:**

Platinum-based chemotherapy and PARP inhibitors are two types of treatment that can benefit prostate cancer patients with mutations in genes that are involved in the repair of damage in the DNA. However, patients who receive one type of treatment may not respond as well to the other treatment later on. The optimal sequencing of these agents is still unclear. In this study, we looked at how 28 prostate cancer patients responded to both treatments. We found that patients generally responded best to the initially given treatment, but still over 40% of patients responded to the second treatment. We also found that the efficacy of the second given treatment was higher in the order of platinum-based chemotherapy first and PARP inhibitor second, compared to the opposite order.

**Abstract:**

Patients with metastatic castration-resistant prostate cancer (mCRPC) harbouring homologous recombination repair-related gene aberrations (HRRm) can derive meaningful benefits from both platinum-based chemotherapy (PlCh) and PARP inhibitors (PARPi). Cross-resistance between these agents is well-recognised in other tumour types but data on prostate cancer is lacking. In this retrospective pre-planned study, we assessed 28 HRRm mCRPC patients who received PlCh and PARPi. Progression-free survival (PFS) on initial therapy was longer than on subsequent therapy (median 5.3 vs. 3.4 months, *p* = 0.016). The median PFS of PlCh was influenced by the order of agents, with 3.6 months shorter PFS after PARPi than when administered first. The median PFS of PARPi was less influenced, with 0.9 months shorter PFS after PlCh than before. In the PARPi-first subgroup, six out of 16 evaluable patients (37.5%) had a >50% PSA decline to PlCh, and two of eight (25.0%) had a radiographic response to PlCh. In the PlCh-first subgroup, 6/10 (60.0%) had a >50% PSA decline, and 5/9 (55.6%) had a radiographic response to PARPi. These data show >40% of the cohort is sensitive to a subsequent HRR-targeting agent. PlCh appears to induce less cross-resistance than PARPi. Additional data on resistance mechanisms will be crucial in defining an optimal treatment sequence in HRRm mCRPC patients.

## 1. Introduction

Treatment is becoming more personalised for patients with metastatic castration-resistant prostate cancer (mCRPC) and loss-of-function alterations in genes involved in DNA damage repair (DDR). Within the DDR pathway, genes associated with homologous recombination repair (HRR) are impaired most frequently. Over a quarter of the mCRPC population harbours at least one pathogenic alteration in an HRR-related gene, most often in *BRCA2* [1,2].

Poly-(ADP)-ribose polymerase inhibitors (PARPi) and platinum-based chemotherapies, as monotherapy or combined with a taxane, (PlCh) show beneficial responses in mCRPC patients with deleterious alterations in genes associated with HRR (HRRm) [1,3,4,5,6,7,8]. Treatment with PARPi or PlCh eventually leads to double-strand breaks, as indicated by DDR through the formation of RAD51 foci, an established functional biomarker of HRR [9,10]. For both therapies, loss-of-function alterations in *BRCA1* or *BRCA2* are associated with the highest rates of response [11,12]. 

The PARP inhibitor olaparib has been approved by both the EMA and the FDA for the treatment of *BRCA*-altered or HRRm mCRPC, respectively, after progression on a 2nd generation AR-axis inhibitor [1]. The PARP inhibitor rucaparib has been approved by the FDA for the treatment of *BRCA*-altered mCRPC patients who have previously received a 2nd generation AR-axis inhibitor and taxane-based chemotherapy [3]. Where PARPi is becoming readily available for HRRm mCRPC patients, PlCh is not considered standard of care for HRRm mCRPC patients, despite accumulating evidence of their effectiveness in *BRCA*-altered patients [7,11,13,14]. Ongoing trials are investigating a role for PlCh, often combined with a taxane, in HRRm prostate cancer (NCT02598895, NCT05461261, NCT02955082).

The optimal treatment sequence of these therapies for HRRm mCRPC is yet to be determined. Cross-resistance between PlCh and PARPi is well described in other tumour types and evidence is emerging on reversion mutations in HRR genes of mCRPC patients following treatment with PARPi or PlCh [15,16,17,18,19,20]. However, data on cross-resistance between PlCh and PARPi in mCRPC is lacking. This pre-planned retrospective study describes responses to PlCh and PARPi in HRRm mCRPC patients who received both therapies. Based on a limited number of patient data from mCRPC, extrapolation from other tumour types, and biological data on cross-resistance mechanisms, we hypothesize that PARPi will induce more cross-resistance towards PlCh, than PlCh will towards PARPi [11]. 

## 2. Materials and Methods

All mCRPC patients from the outpatient clinic for Medical Oncology and/or Urology at the Radboudumc treated with both a PARPi and PlCh were included. Patients were identified by systematic analysis of all PARPi-treated patients for also having received PlCh. Eight patients (study subjects: 7, 9, 10, 12–14, 16, 22) were previously described [11]. Patient follow-up was until 1 December 2022. Efficacy was assessed by the order of treatment; initial treatment with a PARPi followed by subsequent PlCh (PARPi >> PlCh) or vice versa (PlCh >> PARPi). The order in which the agents were administered depended on the physician’s choice, which may have been influenced by the reimbursement of PARPi in the first quarter of 2021 in the Netherlands.

Patients had targeted or whole-genome sequencing performed on archived or fresh tumour material by a non-profit service provider or custom in-house platform. The pathogenicity of gene alterations was assessed within the molecular tumour board of the Radboudumc, largely based on guidelines from the American College of Medical Genetics and Genomics and the Association for Molecular Pathology [21,22]. 

Biochemical response was evaluated according to Prostate Cancer Clinical Trials Working Group 3 (PCWG3) criteria [23]. Radiographic response was classified according to Response Evaluation Criteria in Solid Tumors (RECIST1.1) and PCWG3 criteria, or consensus for PSMA-PET evaluation [24,25]. Progression-free survival (PFS) was defined as the time from the start of therapy until the first moment of radiographic progression, clinical progression, or death, censoring at the end of follow-up or at the initiation of the next systemic therapy. All time-to-event data were analysed with Cox proportionate hazards models. Statistical tests were performed in R (v4.1.3, Vienna, Austria) with RStudio (v2022.02.1, Boston, MA, USA). A *p*-value < 0.05 was considered significant. 

## 3. Results

### 3.1. Study Population

All 28 mCRPC patients were considered HHRm, most commonly due to a pathogenic alteration in *BRCA2* (75%) (Figure 1). Baseline characteristics are presented in Table 1. Notably, patients who received PlCh >> PARPi more often showed aggressive disease features, characterised by a significantly shorter time to CRPC and a higher incidence of synchronous metastases. In general, patients received a median of three treatment lines in the mCRPC setting before starting their first HRR-targeting agent (PARPi or PlCh) and only one patient was taxane-naïve. The taxane given in conjunction with the PlCh were together considered as an HRR-targeting agent, the taxane itself separately qualified as a prior treatment line. 

### 3.2. HRR-Targeting Agents

Out of the 28 patients, 16 received the HRR-targeting agents in the PARPi >> PlCh sequence and 12 in PlCh >> PARPi sequence. The HRR-targeting agents did not necessarily follow directly after each other. A comprehensive overview of the treatment sequence of all given systemic therapies in the mCRPC setting is illustrated per patient in Figure A1. The types of PARPi prescribed were olaparib (19×), talazoparib (7×) and rucaparib (2×). Two patients received a PARPi twice: Study ID 1 received a second PARPi after early discontinuation of rucaparib due to toxicity and Study ID 6 received olaparib in combination with abiraterone within the PROpel study (NCT03732820) and subsequently as monotherapy. Carboplatin was the only PlCh prescribed for this cohort, in 14 cases this was combined with 20 mg/m^2^ cabazitaxel, in seven cases with 75 mg/m^2^ docetaxel, and it was administered seven times as monotherapy. Seven patients received a rechallenge of carboplatin. 

### 3.3. Progression-Free Survival

The PFS of the initial HRR-targeting agent was longer than the PFS of the subsequent HRR-targeting agent (median 5.3 vs. 3.4 months, hazard ratio 2.12, *p* = 0.016). This was in line with the biochemical PFS (bPFS) with a median of 4.7 months vs. 3.0 months, and a hazard ratio of 1.74, *p* = 0.076 (Figure 2). 

The PFS and bPFS of PARPi and PlCh did not significantly differ between agents, either when given as an initial or as a subsequent agent (Table A1 and Figure A2). However, the PFS of PlCh was much more strongly influenced by the order of the HRR-targeting agents than the PFS of PARPi. The median PFS of PlCh was 3.6 months shorter when following a PARPi than when PlCh was given prior to PARPi administration. This is in contrast to the median PFS of PARPi following PlCh, which was only 0.9 months shorter than when PARPi was administered as the first HRR-targeting agent (Figure 3).

The median bPFS of PlCh following PARPi was 2.4 months shorter than when it was administered first, while the bPFS of PARPi after PlCh was 0.8 months shorter than when administered first (Figure A3). 

### 3.4. Prostate Specific Antigen Response

As an initial HRR-targeting agent, the median PSA change was −81.2% for PARPi and −92.3% for PlCh. As a subsequent agent, the median change was −76.2% for PARPi and −2.1% for PlCh (Figure 3). In total, 12 of the 26 evaluable patients (46.2%) witnessed a PSA50 on the subsequent HRR-targeting agent; six out of 16 (37.5%) in the PARPi >> PlCh sequence and 6/10 (60.0%) in the PlCh >> PARPi sequence (Figure 1). Following a PSA50 on the initial HRR-targeting agent, five out of 11 patients (45.5%) also had a PSA50 on the subsequent agent in the PARPi >> PlCh sequence, and six out of eight (75.0%) in the PlCh >> PARPi sequence. Biochemical evaluation at twelve weeks was hampered due to the short time of the treatment of the subsequent HRR-targeting agent (Figure A3).

### 3.5. Radiographic Response

Seventeen patients had a radiographic evaluation of the subsequent HRR-targeting agent, seven of them (41.2%) had a partial response on the subsequent agent; two out of eight (25.0%) on PlCh in the PARPi >> PlCh sequence, and five out of nine (55.6%) on PARPi in the PlCh >> PARPi sequence (Figure 1). Of the four patients with a partial response on the initial PARPi, one patient (25%) also had a partial response on the subsequently given PlCh. Of the seven patients with a partial response on initial PlCh, four patients (57%) also showed a partial response to subsequent PARPi.

### 3.6. Overall Survival

The overall survival did not differ depending on the sequence of therapies (Figure A4). From the initiation of the first HRR-targeting treatment, the median overall survival was 14.5 months for the PARPi >> PlCh sequence and 13.6 months for the PlCh >> PARPi sequence (hazard ratio 1.52, *p* = 0.305). From the time point, the patients were diagnosed with castration resistance, the median overall survival was 45.0 months for the PARPi >> PlCh sequence and 33.4 months for the PlCh >> PARPi sequence (hazard ratio 1.42, *p* = 0.401; Figure A4). One should be aware that the patients within the PlCh >> PARPi group more often showed aggressive disease features at the initiation of the first HRR-targeting agent, with a shorter time from initiation of androgen deprivation therapy to castration resistance and a higher incidence of synchronous metastases. A high symptomatic burden was one of the reasons to initially treat patients with PlCh over PARPi

## 4. Discussion

In this pre-planned retrospective study, we describe cross-resistance between PlCh and PARPi in mCRPC patients with HRRm. To our knowledge, this is the largest cohort comparing intra-patient responses to HRR-targeting agents following a description of eight patients treated with the PARPi >> PlCh sequence, extracted from the larger cohort of 109 patients treated with PlCh, as described by Mota et al. [7]. Here, we clearly show decreased responsiveness for the subsequently given HRR-targeting agents but nevertheless, responses are still seen in almost half of the cohort. 

In the study of Mota et al., eight patients were described who received the PARPi >> PlCh sequence, with 29% of patients showing a PSA50 to subsequent PlCh, compared to 40% in our cohort. In their cohort one of seven had a radiographic partial response on PlCh after progression on PARPi; for this therapy sequence, two of seven patients had a such response in our cohort.

In our cohort, the PARPi >> PlCh sequence showed worse efficacy than the PlCh >> PARPi sequence for PFS, biochemical, and radiological response. The observed difference can be plausibly attributed to the restoration of the HRR machinery. This can occur under the selective pressure of PARPi or PlCh through several mechanisms, the most common being the emergence of so-called “reversion mutations” that restore protein function [26,27,28]. Circulating tumour DNA analysis from the phase 2 TRITON2 trial demonstrated the prevalence of reversion mutations in *BRCA1* or *BRCA2* after progression on rucaparib [15]. Among 100 patients with post-progression plasma samples available, 39 patients (39%) had at least one reversion mutation in *BRCA1* or *BRCA2*. Notably, 23 of the 100 patients received rucaparib based on a homozygous loss, seemingly impossible to restore. Data from the phase 2 GALAHAD trial, investigating niraparib in HRRm mCRPC patients, showed an even higher prevalence of reversion mutations; at end-of-treatment at least one reversion mutation was detected in 28 of the 33 patients (85%), excluding those with homozygous deletions [17]. The emergence of reversion mutations in response to PlCh in mCRPC and other tumour types has been sparsely described in the literature [20,27,29]. While the mechanism of action of PARPi purely relies on HRR deficiency, PlCh exerts their anti-tumour activity in all cells through the formation of covalent inter- and intrastrand adducts with DNA [10,30]. As a results, PlCh might induce less reversions mutations and, at the same time, be less influenced by the presence of reversion mutations.

One additional contributing factor might be that PlCh was commonly continued until unacceptable adverse effects or quality of life deterioration, as demonstrated by a non-rising PSA in 63% of patients at the termination of PlCh. The discontinuation of PlCh prior to manifest disease progression and drug resistance might defer the development of reversion mutations. Whether this will lead to better responses on subsequent HRR-targeting agents has to be further elucidated in prospective studies with cross-over. 

Although the administration of the agents in this study was sequential rather than concomitant, it is possible that pharmacokinetic interactions contributed to the observed effect. In a phase 1/1b trial involving 77 women treated with olaparib and carboplatin, pharmacokinetic data demonstrated a 49% increase in the clearance of olaparib when administered 24 h after carboplatin, as compared to 24 h before carboplatin [31]. This effect was attributed to intracellular sequestration rather than total drug loss from the environment. However, another study showed that this effect was lost after two days [32]. Since our study revealed that PlCh >> PARPi demonstrated superior results and our agents were not given concomitantly, we do not consider pharmacokinetic interactions to be a relevant explanation. No significant pharmacodynamic interactions were observed in the phase 1/1b study.

The findings from this study suggest that initial treatment with PlCh may induce less cross-resistance to PARPi than PARPi to PlCh. More information on cross-resistance is anticipated from the COBRA trial (NCT04038502) which compares carboplatin and olaparib for mCRPC head-to-head with a cross-over design. Furthermore, in analogue to ovarian cancer, ongoing trials are investigating PARPi maintenance after platinum-based induction therapy in mCRPC (NCT03442556, NCT04288687).

This retrospective study comes with several limitations, first and foremost the sample size being too small to draw definite conclusions. A limited number of patients were identified from a retrospective systemic analysis of PARPi-treated patients, partly because PlCh is not considered standard of care for mCRPC, and both agents often being initiated in end-stage patients. Moreover, mechanisms of cross-resistance could not be identified in this cohort of 28 patients as pre- and post-tumour or liquid biopsies were only available for three patients (Figure A5).

## 5. Conclusions

In this study, we show decreased responsiveness to the subsequently given HRR-targeting agent, suggesting the presence of cross-resistance between PlCh and PARPi. Still, over 40% of the patients in this cohort remained responsive to the sequential therapy. The PARPi >> PlCh sequence seemed to induce more cross-resistance than the PlCh >> PARPi sequence. These findings are of importance when it comes to the recent EMA and FDA approval of PARPi and the emerging interest in platinum compounds for HRRm mCRPC patients. Since both agents are becoming more widely implemented, it is vital to gain insight into the prevalence of cross-resistance and to find the optimal sequence for these therapies. Future data from multiple same-patient (liquid) biopsies are warranted to unravel the mechanisms defining this cross-resistance.

## Figures and Tables

**Figure 1 cancers-15-02814-f001:**
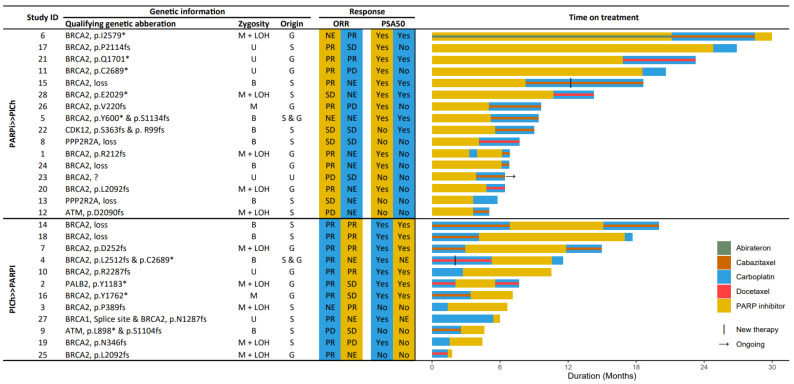
Overview of qualifying genetic aberrations for PARPi treatment, best objective radiographic response (ORR), and PSA decline of >50% (PSA50). Germline regards only the mutation and not the loss of heterozygosity (LOH). The colour of the ORR/PSA50 columns indicates on which HRR-targeting agent the response was (yellow = PARPi, blue = PlCh), the order of the columns is the order as received. The swimmerplot on the right presents the time on treatment on all HRR-targeting agents. B, biallelic; G, germline; M, monoallelic; NE, not evaluable; PARPi, PARP inhibitor; PD, progressive disease; PlCh, platinum-based chemotherapy; PR, partial response; S, somatic; SD, stable disease; U, undetermined.

**Figure 2 cancers-15-02814-f002:**
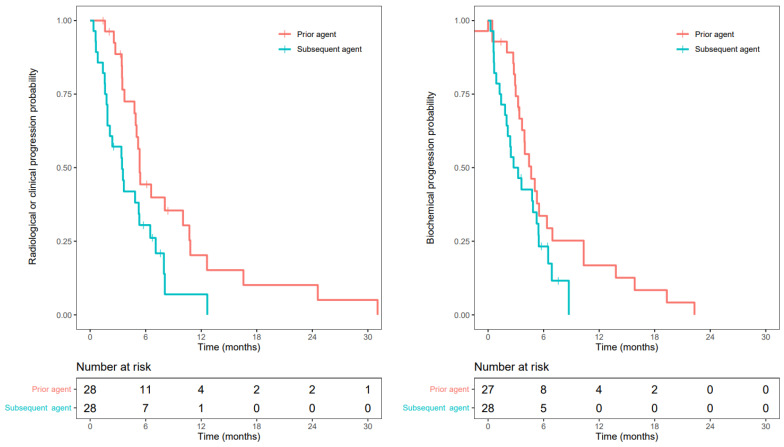
Kaplan-Meier curves with a fraction of patients free from radiological or clinical progression (**left**) or biochemical progression (**right**) for prior and subsequent received HRR-targeting agent.

**Figure 3 cancers-15-02814-f003:**
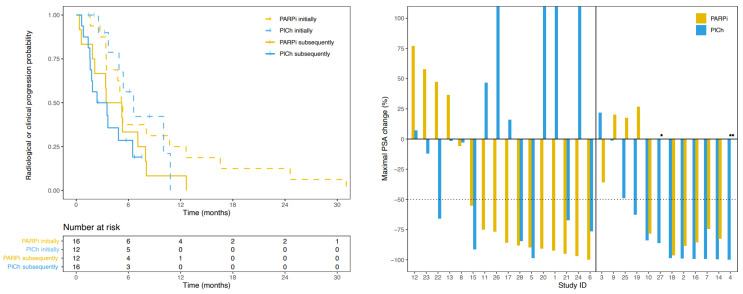
Response on prior and subsequent HRR-targeting agent. (**Left**) Kaplan-Meier curves with a fraction of patients free from radiological or clinical progression by HRR-targeting agent per treatment sequence. (**Right**) waterfall plot of maximal prostate specific antigen (PSA) change on HRR-targeting agents. The order of the bars per study ID represents the order in which the agents were received. * Deceased before first PSA measurement on treatment. ** Baseline PSA < 2 ng/mL.

**Table 1 cancers-15-02814-t001:** Baseline characteristics.

	Total Cohort	PARPi >> PlCh	PlCh >> PARPi	*p*-Value
	Median [IQR] or N (Valid %)			
Age at initial diagnosis, years	61.0 [55.5–64.7]	60.7 [55.5–64.7]	61.3 [55.0–65.5]	0.833
Initial PSA, ng/mL ^1^	91 [11.0–247.0]	68 [15.0–234.0]	98 [9.8–237.0]	0.768 ^2^
Time from start ADT tot CRPC, months	13.5 [8.8–21.9]	17.9 [11.7–25.5]	9.6 [7.3–15.9]	**0.040** ^3^
Age at CRPC, years	62.7 [60.5–66.2]	62.7 [60.5–66.2]	63.1 [56.4–66.2]	0.693
ISUP classification				
1	1 (3.7)	1 (6.7)	0	1.000
2	1 (3.7)	1 (6.7)	0	
3	0	0	0	
4	5 (18.5)	3 (20.0)	2 (16.7)	
5	20 (74.1)	10 (66.7)	10 (83.3)	
Missing	1	1	0	
Synchronous metastasis				
Yes	18 (64.3)	7 (43.8)	11 (91.7)	**0.016**
No	10 (35.7)	9 (56.3)	1 (8.3)	
Treatment before first HRRtA				
Treatment lines for CRPC	3 [2–4]	3 [2–4]	3 [2–3.3]	0.753
Prior taxane	27 (96.4)	15 (93.8)	12 (100)	1.000
Prior ARSI	27 (96.4)	15 (93.8)	12 (100)	1.000

^1^ 3 missing, ^2^ Log-transformation used, ^3^ Square root transformation used. ADT, androgen deprivation therapy; ARSI, androgen receptor signalling inhibitor; CRCP, castration-resistant prostate cancer; HRRtA, homologous recombination repair targeting agent; IQR, interquartile range; ISUP, International Society of Urological Pathology; N, number of patients; PARPi, PARP inhibitor; PlCh, platinum-based chemotherapy; PSA, prostate specific antigen. *p*-values in bold are significant.

## Data Availability

Data are available for bona fide researchers who request it from the authors (https://doi.org/10.17026/dans-2x5-cngv).

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
