# Peer review of "Cross-Resistance between Platinum-Based Chemotherapy and PARP Inhibitors in Castration-Resistant Prostate Cancer"

_cancers, 2023, doi:10.3390/cancers15102814_

Round 1

Reviewer 1 Report

From a clinical epidemiology and oncology biostat point of view, the current pilot research has been very well planned, realized and reported. A small cohort of 28 mCRPC patients received either Platinum-based chemotherapy or PARP inhibitors; the main goal of the Authors was to elucidate potential differences in term of clinical response, following a sequence-dependent administration.

Some comments for the Authors:

- title; I do suggest to modify it as “Cross-resistance between Platinum-based Chemotherapy and PARP Inhibitors in Castration-Resistant Prostate Cancer: a pilot study” just to underline the novelty of sequence-dependent administration

- introduction: have you any pk/pd previous data, concerning this sequence-dependent administration? would you believe that pk/pd infos could help to solve this topic (a short comment on a potential pk/pd interaction has to be added)

- line 78 the order of treatment; how have you decided that, was it depending only on physician decision? This step should be clearly stated

- graphics: simply wonderful swimmerplots, I wish to warmly congratulate the Authors for the big effort in preparing it! Which R packages have been used to prepare them!?

- table 1, you could help the reader adding the number/type of previous therapy lines

- line 121, 16 received PARPi>>PlCh and 12 PlCh>>PARPi; please, see line 78 comment!

- line 128, 75% of carboplatinum pts were also given a taxane chemotherapy, this info should be more deeply underlined all around the manuscript (PARPi vs PlCh plus taxane was the true chemo approach)

- KM curves 2 left and 3 left, the y-axis should be renamed as “radiological or clinical progression probability”

Reviewer 2 Report

Although this is a single institution retrospective study it clearly address some important question in treatment of HRR deficient tumors. Difficencies in the study have been clearly defined by the authors. One point in the difference between use of PARPi and PlCh is the prolonged time on PlCh vs PARP1. Although, the authors note this point, it should be addressed more as to whether it accounts for differences between groups.
